# Performance Analysis of Manufacturing Waste Using SWARA and VIKOR Methods: Evaluation of Turkey within the Scope of the Circular Economy

**Alaeddin Koska and Mehri Banu Erdem \***

Kahramanmaraş Sütçü İmam University, 46000 Kahramanmaras, Turkey; akoska@ksu.edu.tr
\* Correspondence: mbsunbul@ksu.edu.tr

**Abstract:** The increasing population and industrial developments driven by growing needs and expectations have led to an increase in consumption. The rise in consumption, in turn, results in more waste generation. The management of waste has become a global issue concerning human and environmental health. As a solution to climate change, waste, and biodiversity loss, the concept of the circular economy has emerged, which involves a global effort. Zero waste, which is one of the key elements of the circular economy, is regulated by waste management legislation in the European Union in accordance with the waste management hierarchy. Therefore, waste management is an important and urgent issue that requires significant planning, especially for countries with trade relations with the European Union. This study aims to evaluate the performance of waste management in Turkey's manufacturing industry within the scope of the circular economy. The SWARA (Step-wise Weight Assessment Ratio Analysis) and VIKOR (VIseKriterijumsa Optimizacija I Kompromisno Resenje) multi-criteria decision-making methods were used in the research. The examination of manufacturing waste in conjunction with the waste hierarchy and within the scope of the circular economy using multi-criteria decision-making methods sets this study apart from other research on the subject. The analysis results indicate that Turkey, particularly in the preference for the option of selling, has shown an increasing trend in waste reduction, reuse, and recycling indicators, while showing a decreasing trend in disposal. In this context, it can be said that Turkey will not face difficulties in the process of aligning with the European Green Deal, and positive environmental developments have been observed.

**Keywords:** waste management; circular economy; VIKOR; SWARA

## 1. Introduction

After the Industrial Revolution, production techniques based on machine power gained momentum. As a result of mass production, there have been increases in economic growth, improvements in societal welfare levels, and population growth, leading to an increase in production quantities and a subsequent rapid rise in waste generation. In recent years, carbon emissions have occurred due to mass production [1]. Industrial pollution, which is directly related to economic development, necessitates the adoption of environmental management practices by companies to reduce environmentally harmful formations on a global scale [2].

It has become necessary to develop models that protect the environment, preserve biodiversity, and show respect for nature on the path towards sustainable development, without compromising people's living standards. The circular economy is one of these models. With this economic model, which reduces environmental pollution, lowers greenhouse gas emissions, minimizes resource use in economic growth, and creates new job opportunities, it is possible to leave a more livable world for future generations. The primary objective of the circular economy is to restore and regenerate material cycles, meaning to preserve the value of materials throughout a product's lifecycle. This involves minimizing

waste generation and ultimately closing the loop by promoting high-value recycling. The aim is to create a system where materials are continuously reused, recycled, or repurposed, rather than being disposed of as waste. By keeping materials in circulation and extracting a maximum value from them, the circular economy strives to reduce resource depletion and environmental impacts while promoting sustainable economic growth [3,4].

Due to rapid progress, the European Union (EU) economy has outpaced its own production of raw materials. To secure future economic growth, the EU is actively working towards establishing an economy that is sustainable and resource-efficient. This goal is emphasized through the concept of "Closing the Loop", which has been incorporated into EU legislation via the Circular Economy Package. The focus is on minimizing waste generation and promoting its recovery [5]. The practice of waste management is carried out within a procedural framework and is referred to as the waste management hierarchy. With its six components, the waste management hierarchy aims primarily to leave a livable world for future generations. Other objectives of this regulation include creating a sustainable environment, conserving natural resources, saving energy and costs, reducing pollution rates, and minimizing hazardous waste quantities. The waste management hierarchy follows a prioritized order that includes prevention, reduction, reuse, recycling, energy recovery, and disposal [6].

The European Green Deal, in terms of the European Union's (EU) volume of trade within the global trade, is capable of shaping the international trade system. Therefore, it signifies an important transformation process for trading partners, including Turkey. In manufacturing industry establishments, a total of 23.9 million tons of waste was generated, with 4.6 million tons being hazardous. Of the total waste, 56.3% was sold or sent to licensed waste processing facilities, 24.2% was sent to landfill facilities, 7.1% was stored on-site within the workplace, 7% was recovered within the facility, 3.2% was collected by municipal or organized industrial zone (OIZ) authorities, 1.7% was co-incinerated or sent to incineration facilities, 0.4% was used as landfill material or reclaimed by nature, and 0.1% was disposed of through other methods [7]. Considering all the circumstances, what is the current status of Turkey's manufacturing waste in the waste hierarchy and its place in the circular economy? Based on this point, in this study, waste indicators of manufacturing in Turkey have been analyzed using the SWARA and VIKOR methods, and an assessment of the current situation has been conducted within the scope of the circular economy. The manufacturing waste indicators published by the Turkish Statistical Institute [7] were determined as criteria.

## 2. Circular Economy and Waste Management

The European Union (EU), one of the world's largest trade and investment partners, is developing a roadmap for a carbon-neutral economy by 2050 under the framework of the European Green Deal (EGD). One of the key components of the Green Deal, which is recognized as the EU's new economic growth strategy, is the concept of a "circular economy". In 2021 and 2022, regulations have started to be implemented to shape the implementation elements of the EU Circular Economy Action Plan [3]. The Circular Economy is built upon two fundamental cycles: one is biological, and the other is technical [8]. The first component, the biological cycle, involves reducing the excessive exploitation of natural resources, utilizing renewable materials, and reusing organic waste [9]. The technical cycle emphasizes extending a product's lifespan through a hierarchy of circularity strategies, which include reuse, repair, refurbishment, and remanufacturing [10,11]. In summary, within the scope of the circular economy:

- Waste reduction, durability, recycling, reuse, and repair are emphasized as principles of circularity.
- Sustainable product policies are integrated into the system.
- Key objectives include both combating climate change and reducing raw material costs.

Waste, in its simplest form, refers to any material that is generated as a result of production, consumption, and other activities, which is no longer needed and intended to

be discarded [12]. Industrial waste, categorized based on its sources, refers to the waste generated during industrial activities within the framework of production processes. The circular economy is a production and consumption model that encompasses activities such as sharing, leasing, reusing, repairing, refurbishing, and recycling existing materials and products for as long as possible. This approach aims to prolong the lifespan of products, thereby extending their life cycle [13].

The underlying reason for the effort of circularization within the action plan is specifically stated as reducing external dependence, particularly on critical raw materials [9]. In this regard, the EU aims to enhance resource efficiency, recycling, and recovery related to these materials. This expression refers to the waste hierarchy. The waste hierarchy principle has been in existence for about 40 years. It is a concept that emphasizes the prioritization of waste reduction, recycling, and reuse over treatment or disposal methods. The origins of this concept can be traced back to the United States, where the private company 3M initiated it [14].

The current objective of the European Union waste management directives is to encourage waste prevention and the adoption of a waste management hierarchy. However, it is important to note that the Waste Framework Directive primarily focuses on measuring individual waste operations such as recycling, incineration, and landfill [15]. Figure 1 illustrates the waste hierarchy developed for this purpose.

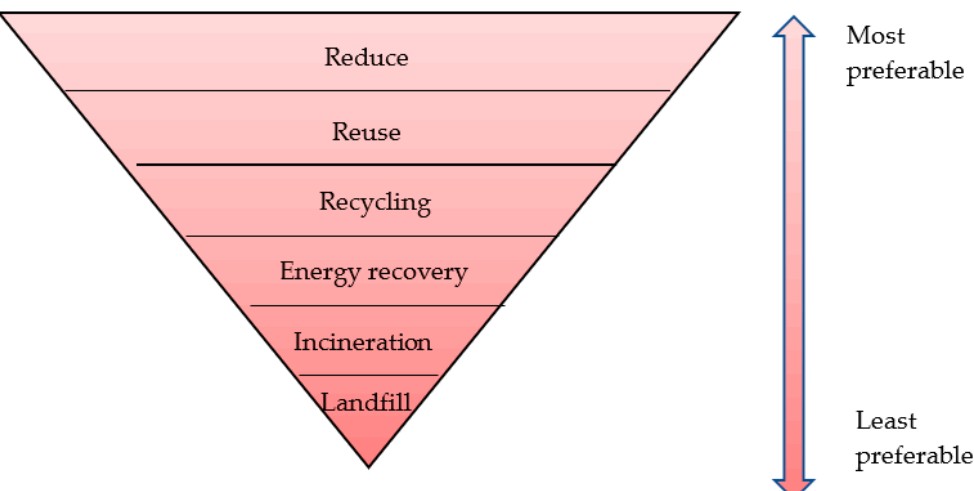

**Figure 1.** Waste Management Hierarchy (Adapted from Zhang et al.) [15].

Hultman and Corvellec [16] support the potential of the waste hierarchy, emphasizing the role of recycling facilities in transforming materials that can be recovered and circulated, which allows for a deeper understanding of material management processes. The 12th SDG is indeed titled "Responsible Consumption and Production" and focuses on promoting sustainable patterns of consumption and production [17]. The specific target related to waste is Target 12.5, which aims to "substantially reduce waste generation through prevention, reduction, recycling, and reuse" by 2030.

Solid waste management is indeed a crucial aspect of addressing climate change and promoting sustainable development in cities, including those in Turkey [18]. The statistics provided highlight the current waste management challenges and potential environmental impacts in the country. The fact that 10 percent of Istanbul's emissions come from the solid waste sector underscores the significance of implementing effective waste management strategies to mitigate climate change. The projected increase in waste generation from 36.4 million tons in 2020 to 58.2 million tons in 2050 emphasizes the need for proactive measures to address this issue. Open dumps are associated with various environmental and health risks, including the release of greenhouse gases, soil and water contamination, and the spread of diseases [7]. The estimated emissions from solid waste in 2020 (27 million

tons of carbon dioxide equivalent) and the expected increase to 35 million tons by 2035 under a BAU scenario highlight the contribution of the waste sector to Turkey's overall emissions profile. Reducing these emissions requires adopting sustainable waste management practices such as waste prevention, recycling, and energy recovery from waste [18]. The data on waste generation from various sectors, including manufacturing, mining, power plants, industrial zones, healthcare facilities, and households, indicate the diverse sources of waste and the importance of addressing waste management comprehensively. The increase in overall waste quantity by 10.5% compared to 2018 further emphasizes the need for improved waste management systems and practices [19].

## 3. Literature

The statement accurately highlights the shift from a linear economic model to a circular economy and its potential benefits for the European system. The linear model, often referred to as the "take-make-dispose" model, is based on a system of continuous resource extraction, production, consumption, and disposal, which is no longer sustainable in the long run [20]. Transitioning to a circular economy entails designing out waste and pollution, keeping products and materials in use for as long as possible, and regenerating natural systems. By adopting this model, Europe can achieve greater resource efficiency, reduce waste generation, and minimize environmental impacts [4]. Some studies on waste management and the circular economy are as follows.

Geyer et al. [21] showcased the current limitations in the potential for reducing the demand for virgin materials through recycling. Van Ewijk and Stegemann [22] argued that the existing waste hierarchy in the European Union, although it gives priority to waste prevention, falls short in achieving a significant reduction in material flow. This inadequacy stems from various factors such as the insufficient specification and implementation of prevention measures, the absence of clear guidance on choosing between hierarchy levels, and the failure to distinguish between open-loop and closed-loop recycling.

Fortuna and Castaldi [23] introduced a reuse indicator known as the Reuse Impact Calculator, which serves to evaluate the influence of reuse organizations on waste prevention within the context of New York City. Taelman et al. [4] sought to develop a conceptual sustainability framework to aid decision-making in waste management within European cities. They examined the urban logistics aspect of waste management within the framework of the circular economy. They emphasized the need for higher-level measures within the scope of eco-design, aiming to prevent or reduce waste generation, extend product lifespan, promote repair, reuse, or integrate recycled materials, in order to reduce the demand for virgin resources.

Pires and Martinho [24] proposed a waste hierarchy index to measure the waste hierarchy within the scope of a circular economy and applied it to municipal solid waste. They distinguished the elements of the waste hierarchy as positive and negative contributors. Recycling and reuse were considered as positive contributors, while incineration and landfill were regarded as negative contributors to the economy. Their approach provided a holistic perspective on how waste is managed. Additionally, Farooque et al. [25] emphasized the significance of conducting additional studies to redirect the focus of waste management towards value recovery.

Redlingshöfer et al. [26] conducted a systematic literature review to demonstrate the limited potential of the waste approach to address the environmental impacts caused by food waste and identified four key insights. These include the waste hierarchy applied to food, assessments for preventing food waste, decision criteria for food waste management, and the waste approach to addressing food waste. Salmenperä et al. [27] examined the critical factors that increase the circular economy in waste management. The study aimed to enhance the understanding of the critical factors encountered by practitioners in transitioning to a circular economy. In this study conducted on industrial waste, they emphasized the need to recognize the interlinkages of barriers and take action at different levels to overcome the obstacles encountered in the transition to a circular economy.

## 4. Materials and Methods

The aim of this study was to evaluate the performance of waste management hierarchy for manufacturing sector wastes in Turkey and assess them within the scope of circular economy. To achieve this objective, the study integrated the SWARA and VIKOR methods, which are multi-criteria decision-making techniques. Every Multiple Criteria Decision Making (MCDM) problem necessitates the selection of weighting methods, as it directly impacts the accuracy and dependability of the decision outcomes [28]. The SWARA method was utilized to obtain the necessary criteria weights for the VIKOR analysis. Additionally, SWARA was used as it helps to ensure a more objective evaluation. The decision-maker identifies the most pertinent criterion based on their perception and subsequently assesses its priority by comparing it with other criteria through ranking [29]. Three environmental engineers were consulted in the SWARA technique. The weights obtained from SWARA were integrated into the VIKOR method for the analysis of waste statistics. The manufacturing waste data were obtained from the Turkish Statistical Institute. The findings obtained from the analyses were interpreted under the waste hierarchy and evaluated within the scope of a circular economy.

### 4.1. SWARA

The SWARA method, which is one of the multi-criteria decision-making methods, was initially proposed by Keršuliene et al. [30]. SWARA stands for "Step-wise Weight Assessment Ratio Analysis" and provides the decision-maker with the freedom to respond without presenting a scale. After asking the decision-maker to rank the criteria, they were asked to compare them with each other. The method was applied separately for each decision-maker, and the obtained data were combined and analyzed.

The steps of the method are as follows: First, the criteria are determined. Then, the decision-makers rank the identified criteria from the most important to the least important. Based on this ranking, the relative importance level of each criterion is determined. For this purpose, it is determined how much more important the jth factor is compared to the $(j + 1)$th factor. This value is expressed as sj by Keršuliene et al. [30]. In the third step, the $k_j$ coefficient is calculated. This coefficient is calculated as shown in Equation (1).

$$k_j = \begin{cases} 1 & j = 1 \\ s_j + 1 & j > 1 \end{cases} \tag{1}$$

As the fourth step, the $q_j$ variable is calculated. The $q_j$ variable is calculated as expressed in Equation (2).

$$q_j = \begin{cases} 1 & j = 1 \\ \frac{q_{j-1}}{k_j} & j > 1 \end{cases} \tag{2}$$

As the final step, the weights of the evaluation criteria, represented by the wj value, are calculated. This calculation process is performed according to Equation (3).

$$w_j = \frac{q_j}{\sum_{k=1}^{n} q_k} \tag{3}$$

### 4.2. VIKOR

The VIKOR method is a decision-making approach proposed by Opricovic and Tzeng [31] for solving multi-criteria problems in complex systems, where conflicting criteria are involved. This method allows for the evaluation of multiple criteria together, generating feasible solutions that are closest to the ideal solution and enabling the selection or ranking of the best alternative based on their performances. The main reasons for choosing the VIKOR method in decision-making problems are its ease of understanding and applicability, as well as its ability to produce realistic solutions. In the context of evaluating each alternative based on individual criterion functions, the compromise ranking can be established by comparing the degree of proximity to the ideal alternative [32].

The foundation of compromise ranking for multi-criteria measurement is based on the $L_p$ criterion used as an aggregation function in compromise programming. Given $J$ alternatives expressed as $a_1,a_2,\ldots,a_j$, the evaluation result for alternative $a_j$ is expressed based on criterion $i$. The $L_p$ criterion, which forms the basis of the VIKOR method, is formulated as:

$$L_P = \left\{ \sum_{=1}^{n} \left[ w_i \frac{f_i^* - f_{ij}}{f_i^j - fi^-} \right]^p \right\}^{1/p} \quad 1 \leq p \leq \infty \quad j = 1,2,3,\ldots,J \tag{4}$$

"$n$" represents the number of criteria here. The $L_{p,i}$ measure given in Equation (4) was introduced by Duckstein and Opricovic and it indicates the distance of $A_i$ from the ideal solution [33]. The common compromise solution $F^c = (f_1^c,\ldots,f_m^c)$ is the closest possible solution to the ideal $F^c$. Compromise refers to an agreement established through mutual concessions represented by $\Delta f_1 = f_1^* - f_1^c$ and $\Delta f_2 = f_2^* - f_2^c$ in Figure 2.

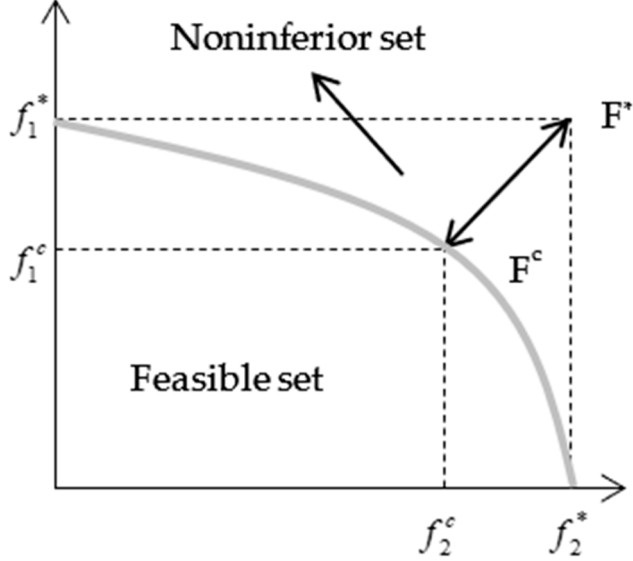

**Figure 2.** Ideal and compromise solutions (Duckstein and Opricovic, 1980) [33].

The VIKOR process steps are as follows.

Step 1: Creation of the decision matrix. The decision matrix ($X$) is a matrix created by decision-makers at the beginning of the process. The rows of the decision matrix represent decision alternatives, and the columns represent evaluation factors. The decision matrix is shown in Equation (5), where $i$ represents the decision alternatives ($i = 1,\ldots,m$) and $j$ represents the evaluation criteria ($j = 1,\ldots,n$).

$$X_{ij} = \begin{pmatrix} x_{11} & \cdots & x_{1n} \\ \vdots & \ddots & \vdots \\ x_{m1} & \cdots & x_{mn} \end{pmatrix} \tag{5}$$

Step 2: The best ($f_j^*$) and worst ($f_j^-$) values are determined for each evaluation critrion. The value that $f_j^*$ and $f_j^-$ will take depends on whether the criterion is of cost or benefit type.

$$f_j^* = \max f_i^j \tag{6}$$

$$f_j^i = \min f_{ij} \tag{7}$$

Step 3: The calculation of $S_i$ and $R_i$ values.

$$S_i = \sum_{j=1}^{n} w_j (f_j^* - x_{ij}) / (f_j^* - f_j^-) \tag{8}$$

$$R_i = \max_j \left[ w_j (f_j^* - x_{ij}) / (f_j^* - f_j^-) \right] \tag{9}$$

$w_j$ represents the criterion weights, indicating their relative importance.

Step 4: The calculation of $Q_i$ values.

$$Q_i = \frac{v(S_i - S^*)}{(S^- - S^*)} + \frac{(1-v)(R_i - R^*)}{(R^- - R^*)}$$
$$S^* = \min_i S_i, \ S^- = \max_i S_i, \ R^* = \min_i R_i, \ R^- = \max_i R_i, \tag{10}$$

$v$ represents the weight for the strategy that maximizes group benefits.

Step 5: Ranking of $S_i$, $R_i$ and $Q_i$ parameters. Creating three ranking lists among the decision alternatives by arranging the values of $S$, $R$, and $Q$ in ascending order.

Step 6: Finding the compromise solution. If the following two conditions are met, option "*a*", which achieves the best ranking in the ascending order of $Q$ values, is proposed as the compromise solution [32].

*Acceptable advantage:*

$$Q(a'') - Q(a') \geq DQ \tag{11}$$

$$DQ = \frac{1}{m-1} \tag{12}$$

The value of option "*a*" in Equation (11) is the second-ranked option in the ascending order of $Q$ values. The parameter 'm' in Equation (12) represents the number of options. If the number of options is less than 4, $D(Q)$ is set to 0.25.

*Acceptable stability in decision-making:* $a'$ should also be the best ranked option in the ranking based on $S$ and/or $R$ values. If one of the conditions is not fulfilled, then the agreed set of common solutions is suggested as follows.

If condition $C_1$ cannot be satisfied, the agreed-upon best solution set is determined to be $a'$, $a''$, $a^m$ and a $(A^1, A^2, \ldots, A^m)$ options. The option a is determined using the formula $Q(a^{(m)}) - Q(a') < DQ$. If the $C_2$ condition cannot be fulfilled, options $a''$ and, that is, the first $(A_1)$ and second $(A_2)$ alternatives, are determined as the best compromise solution.

## 5. Case Study

The manufacturing waste indicators published by the Turkish Statistical Institute [7] were determined as criteria. The data cover the years 2000, 2004, 2008, 2012, 2014, 2016, 2018, and 2020. The inclusion of these years in the evaluation was due to both incomplete data for certain years and the facilitation of the assessment by establishing a specific algorithm. These indicators were grouped into minimum and maximum values based on the value-added stage of activities in the waste hierarchy. These are listed in Table 1 [7].

After determining the criteria, a SWARA evaluation was conducted with the aim of objectively determining the weights to be used in VIKOR. Three decision-makers were selected for the SWARA technique. The steps of the SWARA method are as follows.

The average weights of the indicators according to the expert decision-makers were obtained, as shown in Table 2. Based on this, the importance rankings of the indicators by the decision-makers align with the waste hierarchy. Here, only the relative importance of each indicator compared to the others was weighted. Subsequently, these weights were integrated into the VIKOR method for further analysis.

**Table 1.** The manufacturing waste indicators and description.

| Code | Indicator | Description |
|------|-----------|-------------|
| f1 | The recovered on-site | This involves minimizing waste generation as much as possible, separating recyclable waste at the source, and reintroducing valuable waste back into the economy, which means integrating them into the production process. |
| f2 | The waste sold/sent to licensed companies | Off-site recycling—the practice of businesses selling their waste to relevant facilities (such as recycling plants) instead of conducting on-site recycling processes is aimed at reducing the environmental impact caused by these businesses. |
| f3 | The waste used as fill material/reclaimed in nature | Some waste materials are used as filling materials in infrastructure and superstructure constructions (such as roads, sidewalks, sewage, etc.) with the condition of meeting specific standards for the purpose of recycling. |
| f4 | Co-incineration/burnt in an incineration facility together | This is the combustion of waste as a primary or supplementary fuel in industrial plants and thermal power plants. The conversion of non-recyclable waste, which cannot be recycled or recovered, into usable heat, electricity, or fuel through processes such as anaerobic digestion, incineration, gasification, pyrolysis, and landfilling, is called energy recovery. |
| f5 | Collected by municipality/industrial zones | This is the collection of accumulated waste in front of businesses by the municipality using waste collection vehicles at regular intervals. |
| f6 | Waste sent to sanitary landfill facilities | These areas refer to landfill sites where waste is systematically spread and compacted, and then covered daily. It is necessary for these areas to be meticulously selected and prepared, and for leachate, stormwater, and landfill gas to be controlled. Sanitary landfill facilities are engineered structures designed and operated to dispose of waste while minimizing its impact on public health and the environment. |
| f7 | Stored on the workplace premises | These are storage facilities located within the operation where only the thermal power plant ashes, process residues, and similar wastes generated on-site are disposed of in a liquid form, with the condition of not accepting external waste. |
| f8 | Disposed of by other methods | This is the process of disposing non-recyclable waste, which should be considered as the last option. If not carried out with necessary precautions, it can have negative impacts on the environment. |

**Table 2.** The averages of manufacturing waste indicators.

| Indicator | Mean | Geometric Mean |
|---|---|---|
| The recovered on-site | 0.406 | 0.404 |
| The waste sold/sent to licensed companies | 0.147 | 0.141 |
| The waste used as fill material/reclaimed in nature | 0.188 | 0.185 |
| Co-incineration/burnt in an incineration facility together | 0.120 | 0.109 |
| Collected by municipality/industrial zones | 0.050 | 0.047 |
| Waste sent to sanitary landfill facilities | 0.032 | 0.031 |
| Stored on the workplace premises | 0.035 | 0.035 |
| Disposed of by other methods | 0.022 | 0.022 |

The weights of the indicators are graphically represented in Figure 3. According to this, the recycling of waste within the facility stands out as the most important indicator, significantly surpassing the others. The experts' assessment emphasizes the importance of waste recycling within the facility due to its ability to reduce logistical processes and enable faster entry into the recycling process. This approach not only reduces environmental harm but also highlights on-site recycling as both a priority and a crucial step in waste management. There are two noteworthy observations in Figure 1. Firstly, the indicator of "Sold/Sent to licensed companies" comes closely after the indicator of "Used/Reclaimed as fill material". Secondly, the indicator of "Stored on-site" slightly precedes the indicator of "Sent to controlled landfill facilities". These differences can be attributed to a decision-maker's consideration of the storage facilities available, their suitability, and the organization's ability to manage the process.

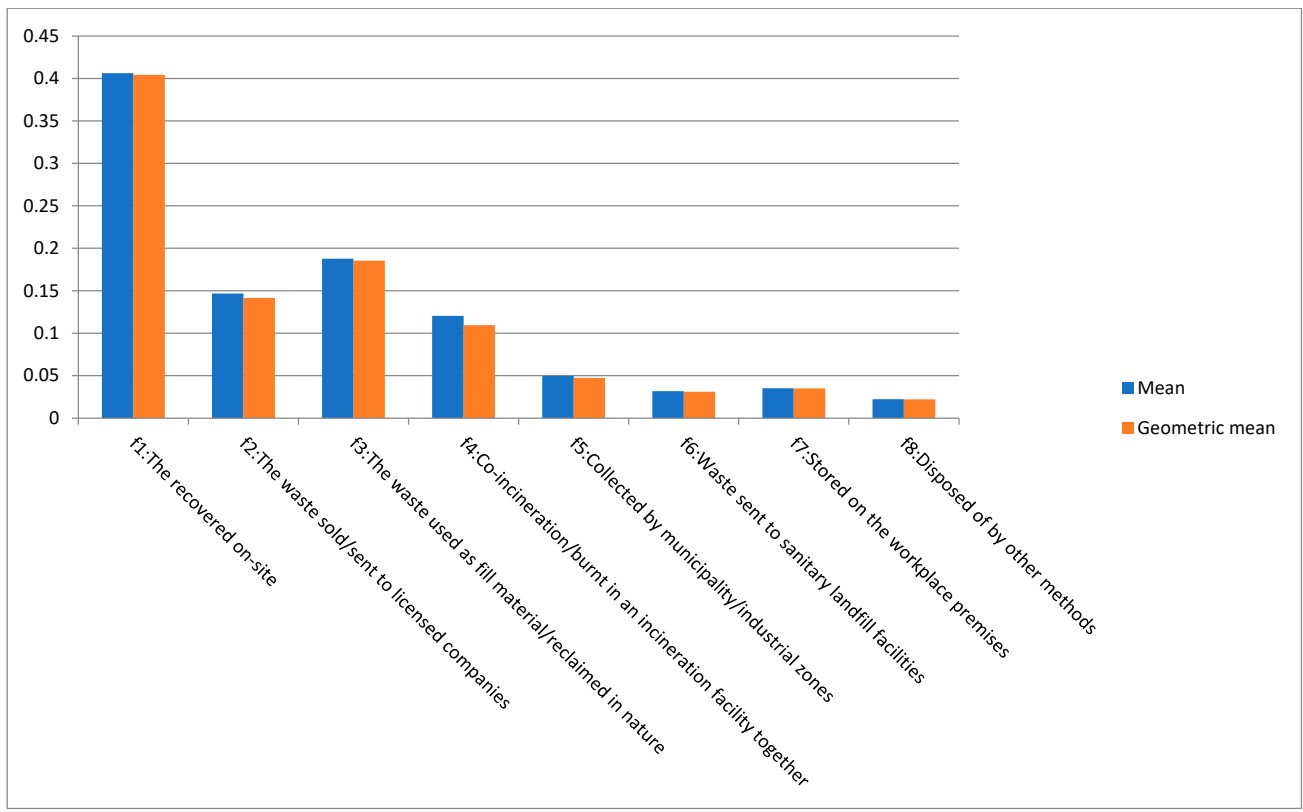

**Figure 3.** The averages of manufacturing waste indicators.

Using the weights obtained through the SWARA method, the analyses of the VIKOR method were conducted. The best and worst values of the indicators were determined

based on the increasing and decreasing order according to the waste hierarchy shown in Figure 1. Furthermore, the study conducted by Pires and Martinho [24] has also been taken into account in this context. They divided the elements of the waste hierarchy into positive and negative contributions. The positive contributions include waste that is recycled on-site, waste sold/sent to licensed companies, waste used as fill material/reclaimed in nature, and waste co-incinerated/burnt in an incineration facility. On the other hand, the negative contributions include waste collected by the municipality/industrial zones, waste sent to sanitary landfill facilities, waste stored on the workplace premises, and waste disposed of by other methods.

In calculating the best and worst values, Equality (6) was used for criteria with a benefit property, while Equality (7) was used for criteria with a cost property. After the operations performed in Excel, the values $f_i^*$ and $f_i^-$ are shown in Table 3.

**Table 3.** Determination of the best and worst criterion values.

| | Indicators | | | | | | | |
|---|---|---|---|---|---|---|---|---|
| | **f1** | **f2** | **f3** | **f4** | **f5** | **f6** | **f7** | **f8** |
| **Years** | Percentage | pct | pct | pct | pct | pct | pct | pct |
| | **Max** | **Max** | **Max** | **Max** | **Min** | **Min** | **Min** | **Min** |
| **2000** | 8.6 | 34.7 | 15.5 | 0.2 | 7.3 | 1.8 | 12.5 | 19.6 |
| **2004** | 7.7 | 45.4 | 3.6 | 1.1 | 9.4 | 5.1 | 4.4 | 23.3 |
| **2008** | 4.9 | 36.5 | 25.6 | 1.8 | 7.8 | 7.4 | 14.8 | 1.3 |
| **2012** | 5 | 43.3 | 1.5 | 1.1 | 3.8 | 33.5 | 10.8 | 1.1 |
| **2014** | 5.4 | 45.1 | 0.9 | 1.3 | 4.3 | 31.1 | 11.8 | 0.1 |
| **2016** | 11.9 | 55.1 | 0.7 | 2.9 | 3.7 | 14.2 | 11.4 | 0.1 |
| **2018** | 9.2 | 57.3 | 0.4 | 2 | 4.5 | 21 | 5.5 | 0.2 |
| **2020** | 7.0 | 56.3 | 0.4 | 1.7 | 3.2 | 24.2 | 7.1 | 0.1 |
| $f_i^*$ | 11.88 | 57.3 | 25.6 | 2.9 | 3.2 | 1.8 | 4.4 | 0.1 |
| $f_i^-$ | 4.92 | 34. 7 | 0.4 | 0.2 | 9.4 | 33.5 | 14.8 | 23.3 |
| **Weights** | 0.41 | 0.15 | 0.19 | 0.12 | 0.05 | 0.03 | 0.04 | 0.02 |

In calculating the values of $S_j$ and $R_j$ representing the average and worst group values for each year, Equations (8) and (9) were utilized. These values are presented in Table 4. After calculating the parameters S\*, S$^-$, R\*, R$^-$ in the step of calculating the $Q_j$ values, the $Q_j$ values were calculated using Equation (10) for different group benefit values based on the parameter $q$ = 0.00, 0.25, 0.50, 0.75, 1.00. The $Q_j$ values are shown in Table 4.

**Table 4.** The calculated values of $S_j$, $R_j$, and $Q_j$.

| | f1 | f2 | f3 | f4 | f5 | f6 | f7 | f8 | $S_j$ | $R_j$ | $Q_j$ |
|---|---|---|---|---|---|---|---|---|---|---|---|
| **2000** | 0.194 | 0.147 | 0.075 | 0.120 | 0.033 | 0.000 | 0.027 | 0.019 | 0.615 | 0.194 | 0.740 |
| **2004** | 0.244 | 0.077 | 0.000 | 0.077 | 0.050 | 0.003 | 0.000 | 0.022 | 0.474 | 0.244 | 0.554 |
| **2008** | 0.406 | 0.135 | 0.000 | 0.047 | 0.037 | 0.006 | 0.035 | 0.001 | 0.667 | 0.406 | 0.809 |
| **2012** | 0.403 | 0.091 | 0.180 | 0.079 | 0.005 | 0.032 | 0.022 | 0.001 | 0.811 | 0.403 | 1.000 |
| **2014** | 0.376 | 0.079 | 0.184 | 0.071 | 0.009 | 0.029 | 0.025 | 0.000 | 0.773 | 0.376 | 0.949 |
| **2016** | 0.000 | 0.014 | 0.000 | 0.000 | 0.004 | 0.012 | 0.024 | 0.000 | 0.054 | 0.024 | 0.000 |
| **2018** | 0.157 | 0.000 | 0.000 | 0.038 | 0.010 | 0.019 | 0.004 | 0.000 | 0.229 | 0.157 | 0.231 |
| **2020** | 0.283 | 0.006 | 0.000 | 0.054 | 0.000 | 0.023 | 0.009 | 0.000 | 0.375 | 0.283 | 0.147 |
| | | | | | | | | * | 0.054 | 0.024 | 0.000 |
| | | | | | | | | - | 0.811 | 0.406 | 1.000 |

Note: * Represents the value of benefit.

After calculating the $Q$ values for each year, these values were used to rank all the years. To determine the stability of the consensus solution in the obtained ranking, the conditions were checked. In this context, the operations performed in Excel and the ranking results are shown in Table 5.

**Table 5.** Verification of conditions.

| a | 0 | 0.25 | 0.5 | 0.75 | 1 |
|---|---|---|---|---|---|
| **2000** | 0.444 | 0.518 | 0.592 | 0.941 | 0.740 |
| **2004** | 0.576 | 0.571 | 0.565 | 0.710 | 0.554 |
| **2008** | 1.000 | 0.952 | 0.905 | 1.027 | 0.809 |
| **2012** | 0.992 | 0.994 | 0.996 | 1.265 | 1.000 |
| **2014** | 0.920 | 0.927 | 0.935 | 1.202 | 0.949 |
| **2016** | 0.000 | 0.000 | 0.000 | 0.020 | 0.000 |
| **2018** | 0.350 | 0.320 | 0.290 | 0.307 | 0.231 |
| **2020** | 0.678 | 0.614 | 0.550 | 0.547 | 0.147 |
| $\mathbf{QA_2}$ | 0.350 | 0.320 | 0.290 | 0.307 | 0.147 |
| $\mathbf{QA_1}$ | 0.000 | 0 | 0 | 0.020 | 0 |
| $\mathbf{QA_2 - QA_1 > 0.143}$ | 0.350 | 0.320 | 0.290 | 0.287 | 0.147 |

According to the VIKOR ranking shown in Table 6, the year 2016 is the best-performing year in terms of the evaluated indicators. On the other hand, 2012 is the worst-performing year. The last three years considered in the ranking (2018, 2020, 2016) demonstrate better performance compared to other years. Prior to 2016, there are slight variations in waste management performance among the earlier years. It is worth noting that the performance of the year 2012 is lower compared to the oldest years in the evaluation, namely, 2000 and 2004.

**Table 6.** Ranking of the years.

| Ranking | $Q_j$ |
|---|---|
| 2016 | 0.000 |
| 2020 | 0.147 |
| 2018 | 0.231 |
| 2004 | 0.554 |
| 2000 | 0.740 |
| 2008 | 0.809 |
| 2014 | 0.949 |
| 2012 | 1.000 |

## 6. Results

According to the findings, while the year 2004 was in a better condition compared to 2000, there was a decline in performance after 2004, which can be attributed to the global financial crisis. The subsequent improvement in 2016 can be associated with increased awareness and efforts towards taking preventive measures following the Paris Climate Agreement, which was published in 2015. Upon detailed examination by year, it is observed that the option of selling waste and sending it to licensed companies stands out more in Turkey.

When looking at the graphs in Figure 4, it can be observed that the disposal option for manufacturing waste ranked second in 2000 and 2004, but its usage decreased after 2004. We can attribute this to the impact of technological advancements in waste management and the development of environmental awareness. Another notable observation in the graphs is the significant increase in the "f3: Used as filling material/reclaimed in nature" indicator in 2008, which was found to occur predominantly in that year. This can be

attributed to the intensified construction activities such as road and railway infrastructure projects, the opening of the Bolu Mountain Tunnel, and the Eskişehir–Ankara high-speed train during those years in Turkey. It is known that using waste materials as filling material in bridges, roads, and construction projects is a common method. Furthermore, it can be seen that the option of using waste as a filling material does not prominently stand out in other years.

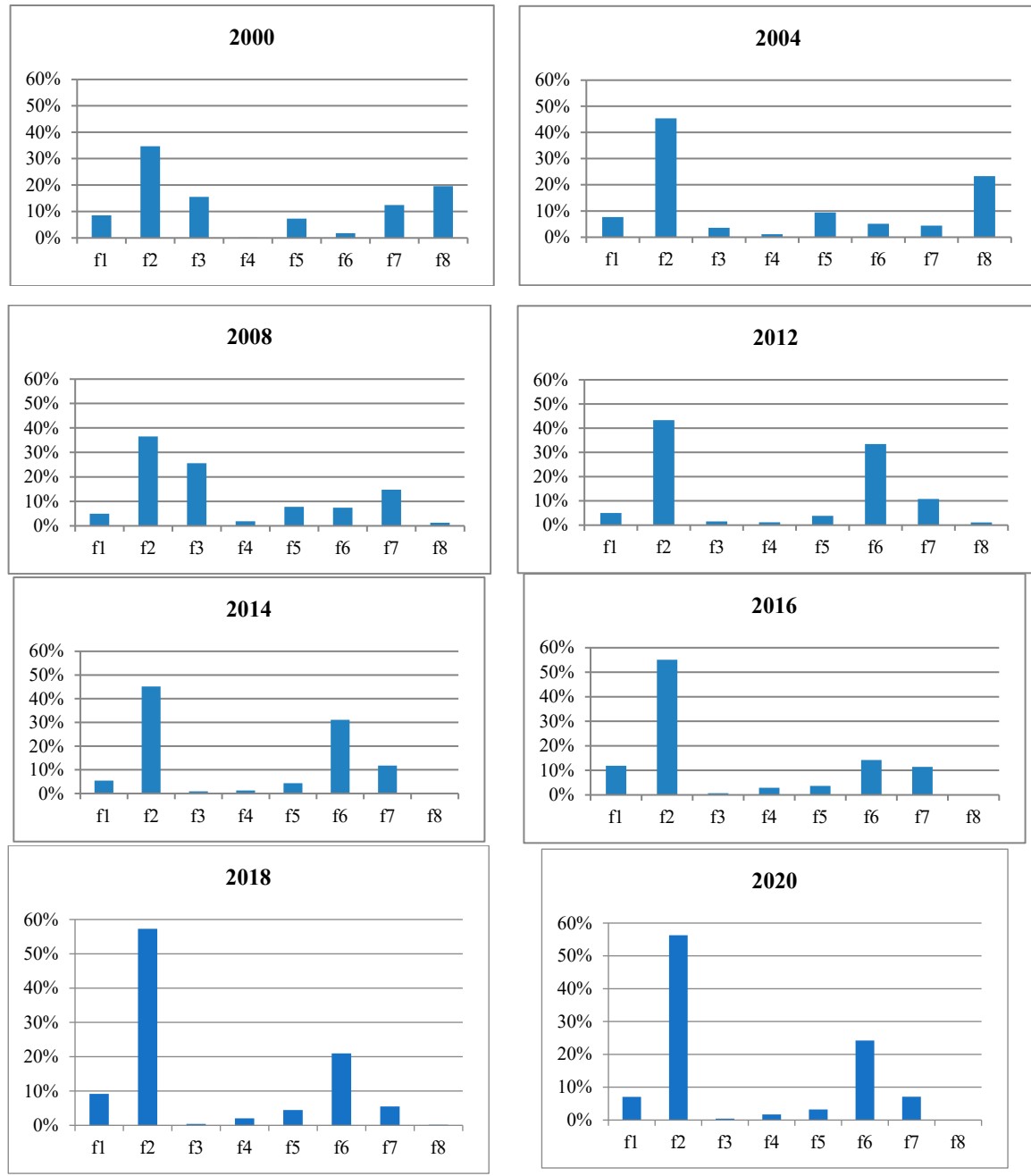

**Figure 4.** Evaluation of Turkey's manufacturing waste by year.

The ratio of f6: waste sent to controlled disposal facilities, and f7: waste stored in workplace premises has increased in relation to the generated waste quantity as of 2008. Particularly, the option of sending waste to controlled disposal facilities (f6) has significantly increased starting from 2012. In 2008, the option of storing waste in workplace premises was preferred over sending it to disposal facilities. However, as observed in the graphs, overall, selling waste is the prominent option in Turkey. The prominence of this indicator suggests

that the number of waste storage sites and recycling facilities is insufficient compared to the generated waste quantity.

Until 2008, the option of f4, "Co-incineration/Burning in incineration plants", was somewhat preferred, but its preference rate significantly decreased after 2008. The option of co-incineration/burning in incineration plants represents energy recovery and is a costly step that requires significant investments. Therefore, it can be said that it is not among the priority choices in developing countries. However, when examining the years in which the option of co-incineration decreased, it can be observed that the quantity of waste "sold to licensed firms" increased. This situation can be attributed to the country's unwillingness to bear the costs of energy recovery options, possibly due to the financial crisis.

Looking at the waste generation in Turkey according to the years in Figure 5, it can be observed that the highest amount of waste was generated in 2020 and 2018, respectively. However, in terms of waste management performance, these years are among the top three. Therefore, it can be stated that Turkey has shown significant developments in environmental activities and waste management in recent years. In this context, it can be predicted that Turkey will achieve the waste management goals set under the circular economy framework.

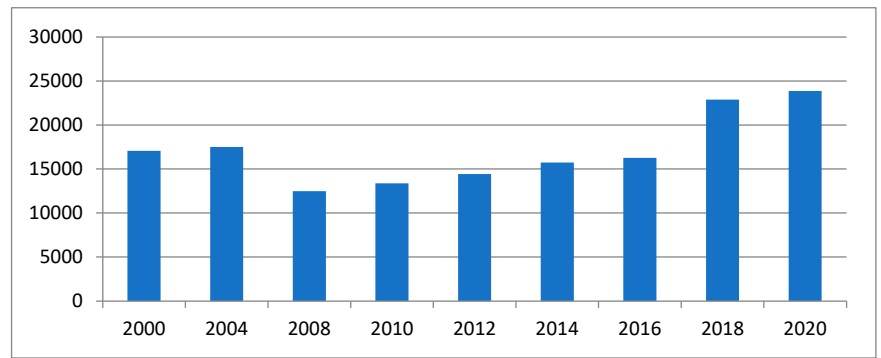

**Figure 5.** Waste generation in Turkey by year (thousand tons) [7].

The higher greenhouse gas emission values in the European Union, which is predominantly composed of industrialized countries, compared to Turkey (Table 7), indicate that Turkey plays an important role in terms of the global impacts of climate change. As shown in Table 7, Turkey's waste values have increased in quantity compared to 1990 in 2018, but their share in total waste has decreased [34].

**Table 7.** Comparison of the sectoral distribution of total greenhouse gas emissions for the years 1990 and 2018 (million tons of $CO_2$ equivalent) [7–10].

| | | 1990 | 2018 | 1990 (Except LULUCF) % | 2018 (Except LULUCF) % |
|---|---|---|---|---|---|
| EU-27 + United Kingdom + Iceland | Waste management | 241 | 138 | 0.3 | 3.3 |
| | Energy | 4350 | 3284 | 76.9 | 77.6 |
| | IPPU | 516 | 374 | 9.1 | 8.8 |
| | Agriculture | 547 | 436 | 9.7 | 10.3 |
| | LULUCF | −245 | −264 | - | - |
| Turkey | Waste management | 11.1 | 17.8 | 5.1 | 3.4 |
| | Energy | 139.6 | 373.1 | 63.6 | 71.6 |
| | IPPU | 22.8 | 65.2 | 10.4 | 12.5 |
| | Agriculture | 45.8 | 64.9 | 20.9 | 12.5 |
| | LULUCF | −55.8 | −95.6 | - | - |

When examining greenhouse gas emissions from the perspective of waste management, which is the main subject of the study, it can be observed that in 1990, waste manage-

ment accounted for approximately 4.3% of total emissions, whereas in 2018, it accounted for 3.3%. Furthermore, when analyzing the share of Turkey's emissions attributed to waste management in the total emissions, it decreased from 5.1% in 1990 to 3.4% in 2018. This indicates that while there has been a decrease in waste management-related emissions both in terms of quantity and proportion in the EU, Turkey has experienced an increase in quantity but a decrease in proportion [28].

When compared to previous similar studies, Şahin and Önder [34] stated in their study that waste-related greenhouse gas emissions in Turkey have been decreasing since 2000, reaching similar results. The decreasing trend in greenhouse gas emissions from waste, despite an increase in waste generation, indicates that waste is being properly managed. Salmenperä [27] supports this study by emphasizing the need for the harmonization of regulations and interpretations, and suggests that the waste management sector can play more diverse roles in implementing the circular economy, such as providing waste processing services for the needs of the manufacturing industry.

## 7. Conclusions

Environmental issues are important global challenges that are on the agenda of the entire world. Therefore, countries are mobilized and actively seeking solutions in this regard. One of these solutions is the circular economy, which is addressed within the framework of the European Green Deal, led by the European Union, and is at the forefront in the fight against climate change. The circular economy aims to increase the efficient use of resources, reduce waste, mitigate environmental pollution and climate change, and preserve biodiversity. In this context, the waste hierarchy becomes a guiding principle. Countries should consider demonstrating a strong presence at the top level of the waste hierarchy and strive to avoid the disposal option as much as possible, viewing it as a duty for all of humanity.

The emergence of the European Green Deal in 2019 and its significance for the Turkish economy serve as the starting point for this study. The aim of the research is to present the current state of waste management in Turkey within the scope of the circular economy. It aims to determine the readiness for the adaptation process on the path towards achieving the carbon reduction targets set for 2030 and 2050.

According to the overall findings of the research, Turkey follows an increasing trend in indicators such as waste reduction, reuse, sale, and recycling, while showing a decreasing trend in terms of disposal. In this context, it can be said that Turkey will not face difficulties in the process of adapting to the European Green Deal. On the other hand, Turkey demonstrates its best performance in the measurement of manufacturing waste management in the years 2016, 2020, and 2018, respectively. Therefore, it can be inferred that significant environmental developments have taken place in recent years, indicating that efforts have been made in this direction.

Developing countries should strive to prevent resource waste, reduce waste generation, minimize waste disposal, and maximize reuse and recycling in their development efforts. In doing so, the country will not only achieve energy savings but also utilize its resources in the most optimal way. Setting strategies and engaging in activities in this direction serve the circular economy.

Turkey should increase the number of waste disposal sites and recycling facilities and establish strategies to reduce waste generation. Minimizing waste is advantageous for a country with insufficient waste disposal and recycling facilities. Therefore, activities related to waste reduction, which are at the top of the waste hierarchy, should always be prioritized. Secondly, planning can be made for the source separation of waste. Source separation results in certain savings in the recycling process. This helps prevent pollution and reduce water consumption for cleaning purposes. To address these challenges, Turkey can focus on several strategies, including: (1) Implementing the waste hierarchy: prioritizing waste prevention, reuse, recycling, and energy recovery over disposal methods such as landfilling or open dumping. (2) Improving waste infrastructure: investing in

waste management infrastructure, including recycling facilities, composting plants, and waste-to-energy facilities, to enable efficient and sustainable waste management. (3) Promoting circular economy practices: encouraging the transition to a circular economy, where waste is minimized, and resources are conserved through strategies like product design for recyclability, extended producer responsibility, and sustainable consumption patterns. (4) Strengthening regulations and enforcement: enhancing waste management regulations, waste segregation practices, and enforcement measures to ensure compliance and proper waste handling. (5) Raising public awareness: educating and engaging the public on waste management practices, promoting behavioral change, and encouraging responsible consumption and waste reduction at the household level. (6) By addressing solid waste management comprehensively and adopting sustainable practices, Turkey can reduce its environmental footprint, contribute to climate change mitigation efforts, and promote a more sustainable future.

Preventing the formation of unconscious waste and reusing unavoidable waste as raw materials, materials, spare parts, etc., should be prioritized. If these activities are not possible, disposal should be carried out without causing harm to the environment. Waste management planning should be carried out accordingly, and waste policies aiming to increase the upper levels and reduce the lower levels of the waste hierarchy should be established. In this regard, a regulatory and supervisory system should be established to control waste in the manufacturing sector. The evaluation of manufacturing waste in terms of the circular economy using multi-criteria decision-making methods demonstrates the significance of this study.

Like any other research, this study also has some limitations. One limitation is that the data are limited to specific years due to the unavailability of data for all years. Another limitation is the restriction in interpreting the data due to the unavailability of sub-sector data. For future research, it is recommended to conduct a similar study by focusing on a specific sector and conducting comparative studies with other countries. Additionally, research can be conducted on other waste sources and types, apart from manufacturing waste. Furthermore, other multi-criteria decision-making methods can be utilized in performance measurements.

**Author Contributions:** Conceptualization, M.B.E. and A.K.; methodology, M.B.E. and A.K.; formal analysis, M.B.E.; investigation, A.K.; resources, M.B.E.; data curation, A.K. All authors have read and agreed to the published version of the manuscript.

**Funding:** This research received no external funding.

**Institutional Review Board Statement:** Not applicable.

**Informed Consent Statement:** Not applicable.

**Data Availability Statement:** The data presented in the article were obtained from the Turkish Statistical Institute. https://data.tuik.gov.tr/Bulten/Index?p=Atik-Istatistikleri-2020-37198#:~:text=Olu%C5%9Fan%20at%C4%B1k%20miktar%C4%B1%20104%2C8%20milyon%20ton%20olarak%20hesapland%C4%B1(1)&text=Toplam%20at%C4%B1k%20miktar%C4%B1%202018'e,9%20milyon%20ton%20at%C4%B1k%20olu%C5%9Ftu, accessed on 20 March 2023.

**Conflicts of Interest:** Our research article titled 'Performance Analysis of Manufacturing Waste Using SWARA and VIKOR Methods: Evaluation of Turkey within the Scope of the Circular Economy' has no financial conflicts of interest with any institution, organization, or individual, and there are no conflict of interest among the authors.

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
