# Peer review of "Performance Analysis of Manufacturing Waste Using SWARA and VIKOR Methods: Evaluation of Turkey within the Scope of the Circular Economy"

_sustainability, doi:10.3390/su151612110_

Round 1
Reviewer 1 Report
Dear Authors
I congratulate you for the research carried out and for the chosen theme written in the manuscript sustainability-2510070 in which you addressed issues related to the circular economy in Turkey, with the itulisation of multicriteria methodology. The authors obtained significant results that justify the relevance of the research. However, we observed some issues that need to be improved, which I present below, before considering the paper for publication:
1) I suggest the authors present the problem question in a direct way in the introduction;
2) introduce a brief summary of the following sections for the reader.
3) between lines 183-193, the authors present the multicriteria tool they used. I request that the authors introduce in the text a justification for choosing and combining these two multi-criteria methods SWARA and VIKOR, to the detriment of countless other methods and combinations. I suggest considering these two papers for this answer since they are two literature reviews on multicriteria methods: https://doi.org/10.3390/info14050285; https://doi.org/10.3390/systems11030125.
4) Inform how the criteria for the model were chosen;
5) Inform why you did not perform sensitivity analysis or comparison with another model to assess the consistency of the ranking;
6) I suggest resizing figure 4, so that it fits on the page of its title.
7) I suggest you review the bibliography entries in the reference section. Read the instructions for authors:
Good revision
Reviewer
Author Response
We would like to express our gratitude to the esteemed reviewer for their valuable assessments that have increased the value of the study. The necessary corrections have been made in the indicated areas. Responses to the reviewer's evaluations are provided in the attached document.

Reviewer 2 Report
Please always spell out the acronym the first time it is used in the body of the paper, for example: VIKOR, SWARA,SDG…
The presentation of the referencing of lines 120 to 137 can be improved, the paragraph is too long for a single reference.
It would be useful if you could specify the VIKOR AND SWARA Methods efficiency in the conclusions.
Funding is not included. If no funding exist, please add: “This research received no external funding”.
The contributions of each author are not included.
Institutional Review Board Statement, Informed Consent Statement are not included. If no exist, the authors should state: " Not applicable ".
You should avoid including bibliography references in the conclusions section, comparative analyzes can be included in the discussion section.
It would be useful if you could specify the novelty in the abstract and the conclusions section.
There is no discussion. In the Results section are presented only the most relevant results obtained. The results obtained through these methods can be compared to other methods currently used with similar characteristics.
Conclusions need to be more precise. The first part of the conclusions seems to be a summary of the document and the final paragraphs correspond to the discussion with other investigations.
Please make sure your conclusions section underscores the scientific value-added of your paper, and the applicability of your results.
Minor editing of English language required
Author Response

(The authors gave the same response as above.)

Reviewer 3 Report
Dear Authors
The topic of the circular economy has become a leading theme for many researchers around the world in recent years. It concerns both the approach and building social awareness, but also ad hoc activities aimed at reducing pollution and the negative impact of plastic waste on the natural environment.
As part of joint actions, an approach aimed at the highest possible degree of recycling and reduction of plastics landfilling should be promoted.
Actions in this direction will be supported by the introduction of new EU directives aimed at verifying and reducing the carbon footprint of manufactured goods.
Best Regards
Author Response
We sincerely thank the esteemed reviewer for their valuable assessments.
Round 2
Reviewer 1 Report
Dear Authors
Congratulations on the extensive review you have carried out on the manuscript sustainability-2510070. I could see that the suggestions the reviewers indicated in the first revision round have been implemented. When reading the new text, I did not notice any other improvements that could be made that would imply a significant increase in the quality of the paper. Therefore, I believe that the manuscript is fit to be considered for publication.
With my compliments.